# Uterine distention as a factor in birth timing: retrospective nationwide cohort study in Sweden

Jonas Bacelis,[1] Julius Juodakis,[2] Kristina M Adams Waldorf,[1,3,4] Verena Sengpiel,[1] Louis J Muglia,[5,6] Ge Zhang,[5,6] Bo Jacobsson[1,7]

GZ and BJ contributed equally.

## ABSTRACT

**Objectives** To determine whether uterine distention is associated with human pregnancy duration in a non-invasive observational setting.

**Design** Retrospective cohort study modelling uterine distention by interaction between maternal height and uterine load.

**Setting** The study is based on the 1990–2013 population data from all delivery units in Sweden.

**Participants** Uncomplicated first pregnancies of healthy Nordic-born mothers with spontaneous onset of labour. Pregnancies were classified as twin (n=2846) or singleton (n=527 868). Singleton pregnancies were further classified as carrying a large for gestational age fetus (LGA, n=24 286) or small for gestational age fetus (SGA, n=33 780).

**Outcome measures** Statistical interaction between maternal height and uterine load categories (twin vs singleton pregnancies, and LGA vs SGA singleton pregnancies), where the outcome is pregnancy duration.

**Results** In all models, statistically significant interaction was found. Mothers carrying twins had 2.9 times larger positive linear effect of maternal height on gestational age than mothers carrying singletons (interaction p=5e−14). Similarly, the effect of maternal height was strongly modulated by the fetal growth rate in singleton pregnancies: the effect size of maternal height on gestational age in LGA pregnancies was 2.1 times larger than that in SGA pregnancies (interaction p<1e−11). Preterm birth OR was 1.4 when the mother was short, and 2.8 when the fetus was extremely large for its gestational age; however, when both risk factors were present together, the OR for preterm birth was larger than expected, 10.2 (interaction p<0.0005).

**Conclusions** Across all classes, maternal height was significantly associated with child's gestational age at birth. Interestingly, in short-statured women with large uterine load (twins, LGA), spontaneous delivery occurred much earlier than expected. The interaction between maternal height, uterine load size and gestational age at birth strongly suggests the effect of uterine distention imposed by fetal growth on birth timing.

## Strengths and limitations of this study

► This is the first study to examine the effect of pregnancy-related uterine distention in a medical register data (ie, non-invasively).

► We used uncomplicated pregnancies from a large, healthy, homogeneous population depleted from risk factors associated with gestational age and adult height.

► To minimise confounding, uterine distention was modelled by two natural phenomena: twin pregnancies and singleton fetuses with very rapid growth.

► The study has implications to the design of future epidemiological, genetic and proteomic preterm birth studies.

► A possible limitation of this study is the assumption that maternal height is a good proxy for uterine size.

For numbered affiliations see end of article.

**Correspondence to**
Mr Jonas Bacelis;
jonas.bacelis@gu.se

## INTRODUCTION

Preterm birth remains the leading cause of adverse outcomes of pregnancy. Globally preterm delivery (PTD) is the leading cause of childhood mortality (under five).[1] Indeed, the long-term health of the child is dependent to a great extent on the length of time spent in the mother's womb as a fetus. Worldwide, PTD rates range from about 5% in some Northern European countries to 18% in Malawi.[2] Preterm labour is a complex phenotype thought to be triggered by several causes including microbial-induced inflammation, maternal stress, uterine distention, decidual haemorrhage and vascular diseases.[3] Complexities in the underlying pathophysiology of the PTD process have been an obstacle to identify effective biomarkers and therapeutics.[4]

Epidemiological studies consistently reported that maternal height (MH) is positively associated with child's gestational age (GA) at birth, which has been replicated in different populations, ethnic groups and time periods.[5–8] Using Mendelian randomisation methods, we demonstrated that this association is likely causal.[9] Building on the causal association observed in uterine distention experiments in primates,[10] we hypothesised that birth is triggered by an interaction between uterine size and constantly increasing

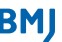

uterine load (UL) (fetal growth). In the current study, we test this hypothesis in humans by modelling the extreme of uterine distention in two natural 'experiments': twin versus singleton pregnancy and in singletons with large and small fetuses for their GA.

## METHODS

### Study population

We investigated pregnancy records in the Swedish Medical Birth Register of mothers who were born in four Nordic countries: Sweden, Norway, Denmark and Finland. Since 1973, it is compulsory for every delivery unit in Sweden to provide the data to this register.

Only the first pregnancy of every mother was used, only with spontaneous onset of labour and only with live-born children. Due to the absence of indicators for spontaneous or iatrogenic onset of labour before the year 1990, only births from 1990 to 2013 were included. Deliveries initiated by elective caesarean section or starting with prelabour rupture of membranes were excluded. Further exclusions contained self-reported maternal medical conditions (diabetes, chronic hypertension, chronic kidney disease, inflammatory bowel disease, systemic lupus erythematosus), pregnancy complications reported in hospital records (placental disorders, placenta previa, early separation of placenta, antepartum haemorrhage, polyhydramnios, oligohydramnios) and estimated maternal body mass index $<15$ or $>45\,kg/m^2$, maternal age $<18$ or $>45$ years, higher order multiple births than twins.

### Variable definitions

Pregnancies with MH values $<140\,cm$ and $>200\,cm$ were removed (0.01%), as well as height values that differed by more than 10 cm from other height values of the same mother in other pregnancies (0.12%). (The latter procedure took place prior to the exclusion of higher-parity pregnancies.)

GA at birth (expressed in days) was estimated by midwives or obstetricians using the second-trimester ultrasonography method and supported by estimated GA at birth reported by the delivery unit.

In order to ensure the initial assumption that twin pregnancies manifest larger UL than singleton pregnancies, we required that birthweight difference of two twins would be no larger than 2 SD of mean intratwin differences in a particular 7-day window of GA (online supplementary figure 1). As a result of this filter, 174 (5.7%) twin pregnancies were removed.

The Medical Birth Register does not collect longitudinal data about the fetal growth (ie, fetal growth curves). However, we used longitudinally derived intrauterine growth curves of healthy Scandinavian pregnancies with term deliveries derived by Marsál et al[11] in order to convert birth weight, fetal sex and GA at birth data into birthweight Z-scores that can be interpreted as being proportional to the fetal growth rate. The newborns with a Z-score larger than 1.5 in the singleton cohort were classified as large for gestational age (LGA), and newborns with a Z-score lower than −1.5 in the singleton cohort were classified as small for gestational age (SGA), online supplementary figure 2.

### Data analysis

We explored how UL modulates the effect of MH on child's GA at birth. We used two models of UL: Model 1 with twin versus singleton pregnancies, and Model 2 with LGA versus SGA pregnancies (all singleton). We assumed that at the time of delivery, twins and LGA singletons manifest a larger volumetric UL than singletons and SGA singletons, respectively.

In both models, we ran multiple linear regression with child's GA at birth being dependent on continuous MH, UL categories and their interaction, where large UL was coded as '1' (twins or LGA) and small UL coded as '0' (singletons or SGA):

$$GA = \beta_0 + \beta_1 \times MH + \beta_2 \times UL + \beta_3 \times MH \times UL + \varepsilon$$

A simple linear regression was preferred to more complex ones (using polynomial terms and covariates) in order to facilitate interpretability of the interaction patterns. The study population, depleted from risk factors associated with GA and height, further justified this choice.

Additionally, we used logistic regression to investigate the same phenomenon in the clinical phenotype—PTD (birth occurring earlier than 259 days of gestation; standard definition). In this analysis, we used extreme dichotomous definitions of maternal stature and UL. Short stature was defined by MH $<161\,cm$ (10th percentile, rounded to integers). In Model 1, high volumetric UL was assigned to twin pregnancies, and in Model 2, high volumetric UL was assigned to singleton pregnancies with extreme fetal growth, defined by fetus having a birthweight Z-score larger than 2.75 SD. Statistical significance was reported as an interaction term p value.

All statistical analyses were performed with R software for statistical computing (V.3.3.1). The analytical code can be found on github.com/PerinatalLab/MatHeight-GestAge/.

### Patient involvement

Patients were not involved in setting the research questions or planning the study. Investigators do not know the identity of study participants.

## RESULTS

Our study, investigating uncomplicated first pregnancies of healthy Nordic-born mothers with spontaneous onset of labour, contained 2846 twin and 527 868 singleton pregnancies. Singleton pregnancies carried 24 286 LGA fetuses and 33 780 SGA fetuses (figure 1). The fractions of excluded pregnancies at each stage are reported in the online supplementary table 1.

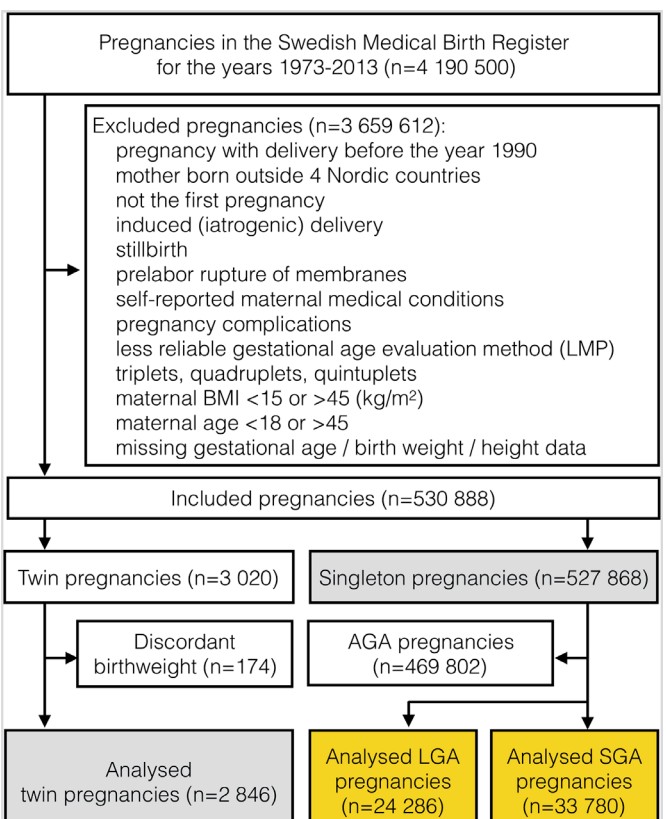

**Figure 1** Flow chart for selection of analysed pregnancies. Two grey boxes represent the analysis in Model 1; two yellow boxes represent analysis in Model 2. AGA, child's birth weight is appropriate for gestational age; BMI, body mass index; LGA and SGA, birth weight is large and small for gestational age, respectively; LMP, last menstrual period.

## Gestational age

In Model 1, we investigated the relationship between child's GA and MH in two categories: women with twin and singleton pregnancies. A clear interaction pattern was detected (figure 2, Panel A): the differing slopes show that the effect of MH on GA is modulated by UL. In particular, the timing of birth was more sensitive to MH in twin pregnancies than in singleton pregnancies. Shorter mothers with twin pregnancies deliver earlier than it would be expected based on the trend in singleton pregnancies and the mean difference between GA in singleton and twin pregnancies. The coefficients estimated in multiple linear regression (table 1) show that MH has three times larger positive linear effect on GA in twin pregnancies ($\beta_3+\beta_1=0.38$ days/cm) than it has in singleton pregnancies ($\beta_1=0.13$ days/cm). The observed effect-modulating behaviour of UL ($\beta_3$ coefficient) is statistically significant (interaction $p=5e{-}14$) and clearly visible in figure 2.

In Model 2, we observed similar results: a clear interaction pattern was observed (figure 2, Panel B), where timing of birth was more sensitive to MH in singleton pregnancies with LGA fetuses than singleton pregnancies with SGA fetuses. Shorter mothers with LGA pregnancies deliver earlier than it would be expected based on the trend in SGA pregnancies and the mean difference between GA in SGA and LGA pregnancies. The coefficients estimated in multiple linear regression (table 2) show that MH has two times larger positive linear effect on GA in LGA pregnancies ($\beta_3+\beta_1=0.2$ days/cm) than it has in SGA pregnancies ($\beta_1=0.1$ days/cm). The observed effect-modulating behaviour of volumetric

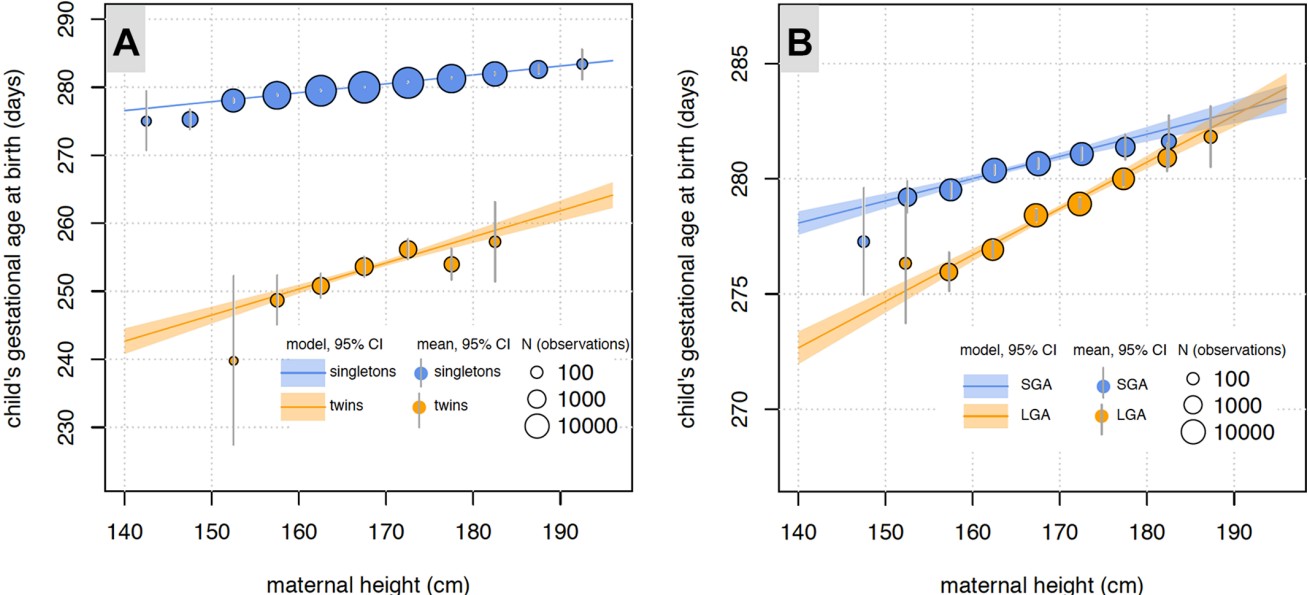

**Figure 2** (Panel A) Interaction pattern in Model 1, where the uterine load is modelled as twin versus singleton pregnancies. (Panel B) Interaction pattern in Model 2, where uterine load is modelled as too large versus too small birth weight for gestational age in singleton pregnancies (large for gestational age (LGA) and small for gestational age (SGA), respectively). Observations are grouped based on maternal height bins. N—count of observations used to estimate mean. The interaction effect of maternal height and uterine load on gestational duration is demonstrated by the different slopes of the two regression lines calculated in the two groups with a different uterine load.

**Table 1** Estimated effect sizes in Model 1 from multiple linear regression

| Coefficient | Estimate | SE | P values |
|---|---|---|---|
| $\beta_0$ (intercept) | 258.2 | 0.412 | <1E−150 |
| $\beta_1$ (MH) | 0.131 | 0.002 | <1E−150 |
| $\beta_2$ (UL) | −69.20 | 5.619 | 8E−35 |
| $\beta_3$ (MH × UL) | 0.252 | 0.033 | 5E−14 |

Estimate is expressed in days of gestation.
MH, maternal height (cm); p values, statistical significance; UL, uterine load (singletons=0, twins=1).

**Table 2** Estimated effect sizes in Model 2 from multiple linear regression

| Coefficient | Estimate | SE | P values |
|---|---|---|---|
| $\beta_0$ (intercept) | 264.6 | 1.646 | <1E−150 |
| $\beta_1$ (MH) | 0.096 | 0.0099 | 5E−22 |
| $\beta_2$ (UL) | −20.13 | 2.598 | 1E−14 |
| $\beta_3$ (MH × UL) | 0.105 | 0.0154 | 1E−11 |

Estimate is expressed in days of gestation.
MH, maternal height (cm); p values, statistical significance; UL, uterine load (SGA=0, LGA=1).

UL ($\beta_3$ coefficient) is statistically significant (interaction p=1e−11) and clearly visible in figure 2.

### Preterm birth

In additional analyses focusing on the clinical phenotype preterm birth, we observed that preterm birth was more prevalent in the risk group of 'short mother and high UL' than it would be expected without synergistic interaction between the two risk factors 'short mother' and 'high UL'.

In Model 1 (figure 3, Panel A), the OR for preterm birth in the 'short mother carrying twins' risk group (Group D, OR=56.2) was larger than it would be expected without interaction between the risk factors 'short mother' and 'twin pregnancy' (OR=1.42×32.8=46.6). However, the interaction term in the logistic regression was not statistically significant (p=0.1).

In Model 2 (figure 3, Panel B), the OR for preterm birth in the 'short mother with extreme fetal growth' risk group (Group D, OR=10.2) was larger than it would be expected without interaction (OR=1.42×2.76=3.92). The corresponding interaction term in the logistic regression was statistically significant (p=0.0005).

### Sensitivity analyses

To address the arbitrary nature of the thresholds chosen to define UL categories in singleton pregnancies, we ran two sensitivity analyses: (1) SGA and LGA defined by birthweight Z-score in ranges (−2.0,−1.0) and (1.0, 2.0), respectively; (2) using continuous UL variable in appropriate for GA singletons with birthweight Z-score in range (−2.0, 2.0). The resulting interaction pattern (online supplementary figures 3 and 4) was qualitatively identical to the results described in the *Gestational age* section.

### DISCUSSION

In this retrospective cohort study, we investigated first pregnancies of healthy Nordic-born mothers delivering

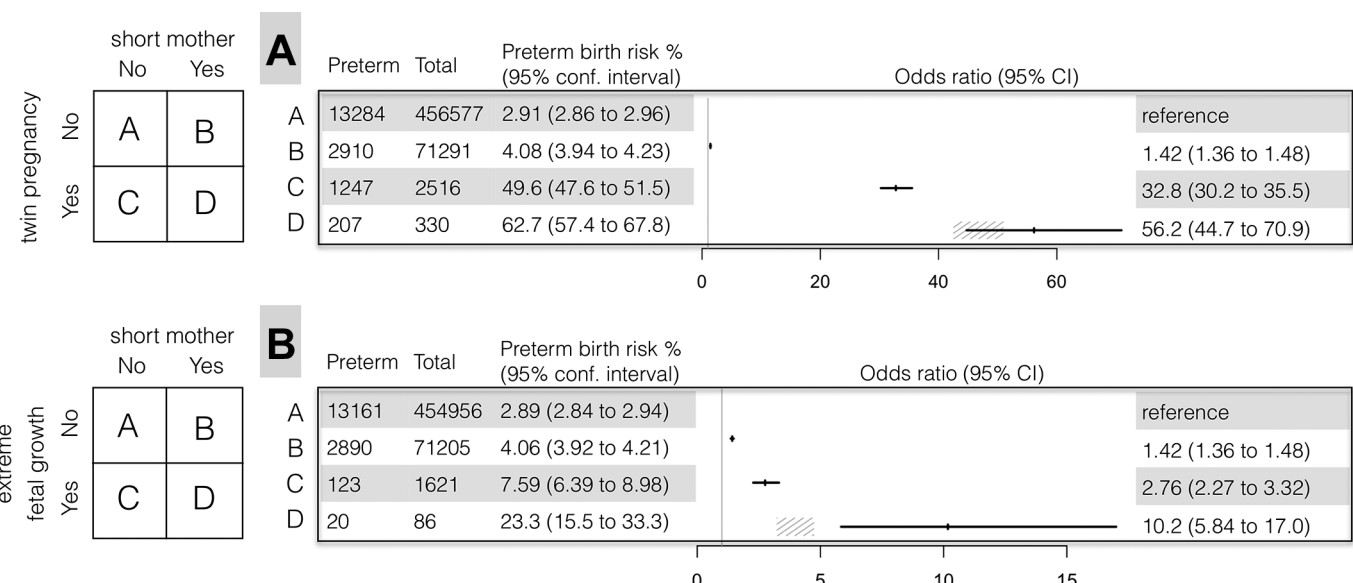

**Figure 3** Preterm birth in short mothers with: (Panel A) twin pregnancies and (Panel B) extreme fetal growth. Short stature is defined as <161 cm. Preterm birth is defined as delivery before 259 days of gestation. Extreme fetal growth is defined as >2.75 birthweight Z-score (adjusted for gestational age). The grey angular lines denote 95% CI for the expected OR in stratum D, estimated as the product of ORs in strata B and C with the assumption of no interaction. In Panel A, the OR in stratum D is larger than expected due to the interaction between maternal height and fetal growth (interaction p=0.1, logistic regression). In Panel B, the OR in stratum D is significantly larger than expected due to the interaction between maternal height and fetal growth (interaction p<0.0005, logistic regression).

in Sweden during the period 1990–2013 with spontaneous onset of labour. We found that child's GA is more sensitive to MH in pregnancies with larger UL than in pregnancies with smaller UL: the effect of MH is significantly larger in twin versus singleton pregnancies, and in singleton pregnancies with fast-growing versus slow-growing fetus. For example, the duration of twin pregnancies is often shorter than of singleton pregnancies. However, in twin pregnancies, a MH increase of 1 cm prolonged pregnancy by 0.38 days, while in singleton pregnancies, a similar increase in MH prolonged pregnancy only by 0.13 days. The observed statistical interaction is likely a manifestation of biomechanical interplay between uterine size and UL (fetal size) in time. The labour in uncomplicated spontaneous delivery is likely triggered by uterine distention, which intensifies with the progression of pregnancy and reaches its peak when fetal size approaches the uterine size.

Our results (interaction pattern) support the previous causality claims for MH[9] and experimental uterine stretch determining the timing of birth,[10 12 13] and suggests that gestation-imposed uterine stretch also affects birth timing in spontaneous human pregnancies. An alternative explanation could be related to metabolic constraint imposed by maternal energy expenditure and fetal energy demand over gestational time,[14] as twins and LGA singletons require more energy to support their growth and maternal basal metabolism is related to height.[15]

A possible limitation of this study is the assumption that MH is a good proxy for uterine size. Similarly, the natural models of uterine distention are not ideal: UL groups also differ in other biological aspects,[16] which cannot be fully accounted for. Another potential limitation is that SGA and LGA were defined using birthweight Z-score, which was estimated using longitudinally derived intrauterine growth curves of healthy Scandinavian pregnancies with term deliveries,[11] that is, not ideally representative population.

The biomechanical explanation of timing of birth could benefit genomic and proteomic association studies of preterm birth by narrowing the number of biomarkers and increasing statistical power: our results could support the use of candidate biomarkers from inflammation-mediated pathways, which are excited by uterine distention.[10 17–20]

The future predictive algorithms could take into account fetal growth speed and uterine size while trying to classify pregnancies as being low or high risk for preterm birth.

## CONCLUSION

The observed interactions support the hypothesis of uterine distention playing a causal role in determining the timing of birth, at the same time giving support to the causal effect of MH, UL and uterine size.

**Author affiliations**
¹Department of Obstetrics and Gynecology, Sahlgrenska University Hospital Östra, Gothenburg, Sweden
²Department of Obstetrics and Gynecology, Institute of Clinical Sciences, Sahlgrenska Academy, University of Gothenburg, Gothenburg, Sweden
³Department of Obstetrics and Gynecology, University of Washington, Seattle, Washington, USA
⁴Department of Global Health, University of Washington, Seattle, Washington, USA
⁵Human Genetics Division, Cincinnati Children's Hospital Medical Center, Cincinnati, Ohio, USA
⁶Center for Prevention of Preterm Birth, Cincinnati Children's Hospital Medical Center, Cincinnati, Ohio, USA
⁷Department of Genetics and Bioinformatics, Area of Health Data and Digitalisation, Norwegian Institute of Public Health, Oslo, Norway

**Acknowledgements** The authors are grateful to The Swedish National Board of Health and Welfare (Socialstyrelsen) which manages the Swedish Medical Birth Register.

**Contributors** JB and GZ: study design and data analysis and interpretation; BJ and VS: data acquisition; JB and JJ: data cleaning and figures; KMAW: literature search; JB and KMAW: writing; BJ, JJ, LJM, VS and GZ: critical revision. All authors approved the final version to be submitted for publication.

**Funding** This work was supported by Swedish government grants to researchers in the public health sector (ALFGBG-717501, ALFGBG-507701, ALFGBG-426411), The Swedish Research Council (2015-02559), March of Dimes Foundation (21-FY16-121) and the Burroughs Wellcome Fund Preterm Birth Research Grant (10172896).

**Competing interests** None declared.

**Patient consent** Not required.

**Ethics approval** The work was approved by the Regional Ethical Review Board in the Western Health Care Region of Sweden (Dnr:968-14).

**Provenance and peer review** Not commissioned; externally peer reviewed.

**Data sharing statement** Researchers can apply for data access by contacting The Swedish National Board of Health and Welfare.

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
