## [Reviewer comments · BMJ Open]

ARTICLE DETAILS

TITLE (PROVISIONAL)	Uterine distention as a factor in birth timing: retrospective nationwide cohort study in Sweden
AUTHORS	Bacelis, Jonas; Juodakis, Julius; Adams Waldorf, Kristina; Sengpiel, Verena; Muglia, Louis; Zhang, Ge; Jacobsson, Bo

VERSION 1 – REVIEW

REVIEWER	Abdool S. Yasseen III Children's Hospital of Eastern Ontario - Research Institute, Ottawa, Ontario, Canada
REVIEW RETURNED	19-Apr-2018

GENERAL COMMENTS	Comments: Methods: - If only the first pregnancy used (page 5 line 6), how is the exclusion of height values >10cm from other height values of the same mother in other pregnancies used (page 5 lines 19-20)? Please expand on this data cleaning step or remove for clarity.- Maternal height, gestational age, twin pregnancies, and fetal growth subheadings should be grouped under a single subheading of “variable definitions”.- The authors make a fair number of exclusions, consisting of elective CS or PROM, self-reported maternal co-morbidities, pregnancy complications, extreme maternal BMI, height, or age, unrealistic birthweights for gestational age, and multifetal pregnancies greater than twins or twins with discordant birthweights.o Please provide a comparison of those excluded through a supplementary table, or at least the frequencies of each exclusion category so that the reader can assess whether the study cohort is representative of the underlying population. This could be inserted into figure 1. At the moment 75% of the underlying population is excluded without mention of the proportion of each reason. It is likely that most of the exclusions are due to the study period restriction (i.e. 1990-2013), but this needs to be stated along with the number of pregnancies within each excluded group.o Also, have the authors considered using a propensity matched design to tighten the regression model? Please comment, in one or two
--

sentences in the data analysis section, why such alternative designs were not considered.

- Linear regression model to assess the interactive effect of maternal height and uterine load on gestational age.
 - o Regression equation on page 7 (line 4) not needed
 - o Small typo on page 7 (line 5) "... a the ...".
 - o Latter part of the second paragraph on page 7 (lines 7-12) should be removed from the data analysis section.
- The interaction model was not absolutely required. Using interaction term in a regression model can conserve statistical power, as compared to the simpler stratified approach, however this usually sacrifices interpretability. It would be better to use a stratified approach, comparing four regression models similar to figure 3. Unless the authors had reason to use a continuous height measure for figure 2 and not for figure 3. Please provide justification of dichotomizing maternal height.
- The simulation models are not required, as they don't contribute to the discussion, and confuse the issue. Please remove.

Results:

- The authors state on page 9 (lines 7-9) that "A clear interaction pattern was detected (figu.2 panel 1)", however, judging from the figure this interactive effect is not evident. Is there a measure of statistical interaction, such as RERI or Synergistic index that the authors could quote to substantiate this claim? Rather, this seems like effect measure modification, as the slopes of the lines are approximately the same with different intercepts. For panel 2, the interactive effect is clear.
- Tables 1 and 2 should not be in the results section text; rather they should be on a separate page each at the end of the manuscript. In fact, both tables could be presented as a single table.
- Model coefficients are statistically significant; however, this is likely due to large sample size, meaning that the models are over powered. Please comment on the magnitude of the coefficient estimate. As an example, for table 1, "the independent effect of maternal height causes a 0.13 days increase in gestational age with every increase in maternal height unit, whereas the independent effect of increased uterine load causes an expected reduction of 69 days." Please rephrase the results section with a focus on the magnitudes, rather than statistically significant p values.
- Please remove simulation analysis results

Discussion:

	 - “iatrogenic uterine stretch” implies that the uterus was stretched by the attending physician or due to treatment. Please correct this sentence. - Please refer to specific study findings when stating “our results support ...”. Which of the study findings support these claims? - Sensitivity analysis mentioned on page 12 (line 18) was never explained in the methods. Also, the study population consisted of first pregnancies, so where did the authors acquire these additional pregnancies? Please remove this sentence or include it in the methods and results. - Please temper the language on page 12 (line 22). “our results implicate candidate genes and proteins from inflammation mediated biochemical pathways...” The results from this study demonstrate an association, which may or may not be used to further research on predicting expected gestational duration.
--	--

REVIEWER	José Derraik • Liggins Institute, University of Auckland, Auckland, New Zealand • Uppsala University, Uppsala, Sweden
REVIEW RETURNED	22-Jul-2018

GENERAL COMMENTS	I have enjoyed reading this interesting and well-written study that aims to examine a likely factor underpinning the previously described inverse association between maternal height and gestational length. This manuscript also highlights once again that the Swedish Birth Register is a powerful and valuable resource. While I think this study is valuable and adds to the literature on this topic, I do have some questions regarding the manner in which the data were handled and analysed. Main concerns:  • In one of its key comparisons, the authors contrast singleton and twin pregnancies. However, I don't think this comparison can be made to assess a possible association between uterine load and pregnancy length. Twin pregnancies are quite different from singleton ones. Most importantly, there is experimental evidence from animal studies showing that the growth trajectory (and to some extent pregnancy length) in twin pregnancies is likely determined very early in gestation (possibly even in the periconceptional period). For example, Hancock et al. showed that the surgical removal of a twin in sheep pregnancy did not lead to birth weight similar to that of singletons, while the length of gestation of such reduction pregnancies more closely approximated that of twin pregnancies rather than that of singletons (Hancock et al. J Physiol 2012;590:1273-84). As a result, I question the validity of singleton and twin pregnancies as surrogate categories for uterine load. • Similarly, I see issues with the comparison between LGA and SGA pregnancies. SGA classifies most pregnancies likely to
---

involve disordered fetal growth, which may result from a wide range of underlying conditions (possibly undiagnosed) (Lee et al. Pediatrics 2003;111:1254-61). In particular, in this cohort solely of Nordic mothers and their offspring, it is unlikely that there were many SGA births resulting strictly from short parental stature. Thus, I think it is reasonable to assume that pregnancies leading to SGA births would unlikely constitute a group of health pregnancies for a valid comparison. On the other hand, a number of LGA births might have also resulted from undiagnosed cases of gestational diabetes mellitus.

- An important factor that the authors do not seem to have accounted for is maternal BMI. There is extensive evidence that maternal obesity is associated with an increased risk of post-term birth (e.g. Heselhurst et al. Obes Rev 2017;18:293-308). Conversely, data from the Swedish Birth Register also showed that maternal obesity is associated with an increased risk of preterm delivery (Cnattingius et al. JAMA 2013;309:2362-70). Thus, maternal BMI is an important confounding factor that needs to be accounted for in the analyses, as it may reduce or magnify the observed associations.

- The authors have adopted some rather arbitrary thresholds without clear justifications for them. These include:
 - non-standard definitions of SGA and LGA (page 6, lines 13-16).
 - the exclusion of mothers with BMI less than 15 kg/m² (may be better to exclude all underweight mothers, i.e. BMI <18.5 kg/m²)
 - what is the rationale for the 2.75 SD threshold? (page 7, line 12)
 - mothers at the upper end of the BMI spectrum were also excluded, but why >45 kg/m²?
 - An intra-individual difference of 10 cm between height measurements is somewhat extreme. I am not sure why such a high threshold was adopted, as it suggests that the measured height of a number of mothers included in the study is likely to be rather inaccurate. This is concerning since maternal height is key for this study.
 - Page 6, lines 1-2 – This statement is vague and needs proper definition as well as adequate justification.

In light of the above, I would suggest tackling the key question differently. For example:

1. Focusing on singleton AGA pregnancies using birth weight SDS as a continuous variable. Also, possibly running stratified analyses, maybe looking at those >-2SD but <-1 SD vs >+1SD but <+2SD

2. While the authors have considered only the first pregnancy for each mother, as the study covers a 15-year period, I think looking at siblings would add considerable value to this study. While it would be necessary to adjust for birth order, it would still be quite interesting to examine whether the reported association persists when one examines gestational length among siblings that were discordant for birth weight. The analyses could also be carried out on twin pregnancies separately.

Other comments:

- Page 5, line 8 – I suggest replacing "kept" with "included".

	 • Page 5, line 10 – Similarly, replace "removed" with "excluded". • Page 7, line 5 – Delete "a" • It is important that the authors include a paragraph commenting on the study's limitations. • The references will also need to be edited to ensure a consistent formatting.
--	--

VERSION 1 – AUTHOR RESPONSE

Reviewers' Comments to Author

Reviewer 1

Reviewer Name: Abdool S. Yasseen III

Institution and Country: Children's Hospital of Eastern Ontario - Research Institute, Ottawa, Ontario, Canada

Competing interests: None declared

Feedback:

Thank you for your assessment of the association between uterine distention and gestational age. This is a wonderful area of research that needs population based data to address research questions. However, as with any population based study, careful thought must be given to who is included and excluded, and the choice of how to analyze the data. My main concern is with the exclusion of a majority of your underlying population, which may introduce severe selection biases. Most women with uterine distention have associated co-morbidities or fetal abnormalities that may be missed by self-reporting or even in the administrative record. In addition, your choice of study design (regression models for assessing interactions) makes interpretation of the results difficult. A stratified approach would read better to a clinical audience. For these reasons as well as those listed in my comments, I suggest that the manuscript in its current version would require major modifications to the analysis and interpretation, before it is ready for publication. I hope these comments are helpful.

Thank you for recognizing the importance of this research and for your kind comments. To directly address your main concern of study exclusions, we repeated the analyses with no exclusions as suggested. As can be seen from this sensitivity analysis, Figure A and Table A below (similar to Figure 2, Panel 2), the qualitative findings are the same; there is a significant difference in the two slopes, one modeling gestational age dependence on maternal height with a large uterine load (LGA) and the other with a small uterine load (SGA). Gestational age is the most sensitive to maternal height in the higher uterine load group. The slope in the SGA group is 0,000 days/cm (no association), the slope in the LGA group is 0,244 days/cm. Similar findings were observed in the sensitivity analysis using twin vs singleton pregnancies (not shown here). Thus, we doubt that the originally applied conservative filters introduced the reported interaction pattern.

As we model uterine distention as a binary variable, our regression model for interaction term is essentially equivalent to a stratified analysis. We demonstrated the interaction patterns (Figure 2)

by the different slopes of the regression lines obtained from each group, which is a stratified approach.

Figure A. Sensitivity analysis. No subjects were excluded in this analysis except twin (and higher order) pregnancies. The large for gestational age group (LGA) is defined as having a birthweight Z-score >1.5 ($n=217,770$), while the small for gestational age group (SGA) is defined as having a birthweight Z-score <1.5 ($n=181,245$).

Table A. Sensitivity analysis.

	Beta	SE	P-value
Intercept	275,1	0,943	$<1E-150$
Maternal height (MH)	0,0003	0,0057	0.96
Uterine load (UL)	-39,09	1,3121	$<1E-150$
Interaction (MH x UL)	0,2438	0,0079	$<1E-150$

Uterine load (UL) was coded as "0" when SGA and "1" when LGA.

We would like to emphasize that the intention of our study was to explore the non-pathological uterine distention imposed by a growing fetus in healthy women with normal pregnancies, which would not include extremes of uterine stretch imposed by abnormalities like polyhydramnios. This was the rationale for excluding complicated pregnancies.

Comments on Methods:

1. If only the first pregnancy used (page 5 line 6), how is the exclusion of height values >10cm from other height values of the same mother in other pregnancies used (page 5 lines 19-20)? Please expand on this data cleaning step or remove for clarity.
This particular data quality control procedure took place prior to the exclusion of pregnancies with higher parities. Now we have made sure that the paragraph explains it clearly.
2. Maternal height, gestational age, twin pregnancies, and fetal growth subheadings should be grouped under a single subheading of “variable definitions”.
Thank you for this excellent suggestion. We now have grouped the subheadings as suggested.
3. The authors make a fair number of exclusions, consisting of elective CS or PROM, self-reported maternal co-morbidities, pregnancy complications, extreme maternal BMI, height, or age, unrealistic birthweights for gestational age, and multifetal pregnancies greater than twins or twins with discordant birthweights.

- Please provide a comparison of those excluded through a supplementary table, or at least the frequencies of each exclusion category so that the reader can assess whether the study cohort is representative of the underlying population. This could be inserted into figure 1. At the moment 75% of the underlying population is excluded without mention of the proportion of each reason. It is likely that most of the exclusions are due to the study period restriction (i.e. 1990-2013), but this needs to be stated along with the number of pregnancies within each excluded group.

Previously, we avoided including such detailed information because many of the applied filters overlap (applying one filter affects the excluded fraction by another filter). Step-by-step summaries can be misleading. However, we agree with the reviewer’s request for clarity; we have now included a detailed order of filters with corresponding fractions of removed samples in the Supplement (Sup Table 1). Even more detailed information can be found in the Github repository online (referenced in the manuscript). This analytical code guarantees full transparency of the applied methods.

- Also, have the authors considered using a propensity matched design to tighten the regression model? Please comment, in one or two sentences in the data analysis section, why such alternative designs were not considered.

Propensity score matching is an excellent technique that would be our first choice in a more straightforward observational study (e.g., investigating exposure-outcome association), with a relatively small sample size, where there is a need to collapse multiple mandatory covariates into a single score. However, our study is none of the above: sample size is large, there are no covariates to collapse, and most importantly, we investigate interaction of three variables, where the use of propensity score matching on both uterine load and maternal height would introduce a great deal of complexity and unpredictable effects that would be difficult to interpret.

To avoid this complexity, we chose not to use covariates, instead limiting the study population to pregnancies that do not have strong gestational-age-affecting risk factors present (e.g., preeclampsia, hypertension) or adult-height-associated risk factors (e.g., nationality, age). This rationale is now included in the manuscript (“Data analysis” section).

4. Linear regression model to assess the interactive effect of maternal height and uterine load on gestational age.
 - Regression equation on page 7 (line 4) not needed
 - Small typo on page 7 (line 5) “... a the ...”.
 - Latter part of the second paragraph on page 7 (lines 7-12) should be removed from the data analysis section.Thank you for your comments. We have now corrected the typo. Regarding the regression equation, we believe that it is necessary to understand the results presented in Table 1.

Regarding the following paragraph (page 7, lines 7-12), we have reviewed this section in detail and prefer to keep it, as it clarifies the ideas to the reader.

5. The interaction model was not absolutely required. Using interaction term in a regression model can conserve statistical power, as compared to the simpler stratified approach, however this usually sacrifices interpretability. It would be better to use a stratified approach, comparing four regression models similar to figure 3. Unless the authors had reason to use a continuous height measure for figure 2 and not for figure 3. Please provide justification of dichotomizing maternal height.

The main reason for choosing an interaction model was that a stratified approach does not provide quantified evidence for the difference between strata. While it is possible to compare the within-strata estimates visually, without an interaction model the reader would be unable to obtain the standard error or significance of these differences - hence it is also not possible to discuss the power of such an approach.

In Figures 2 and 3, we wanted to show the results from two different perspectives. Analyses with continuous height and gestational age (Figure 2) were the most detailed and best-powered to reveal interactions. In contrast, the dichotomized height and gestational age (Figure 3) could present a picture more appealing to a clinical audience; this approach avoided interpretation of interaction beta coefficients, abstained from using descriptors such as "an increase of days-per-centimeter", and referred to easily communicable concepts ("preterm", "short"). We feel that both approaches are helpful to different audiences. The second analysis provides the most intuitive interpretation of interaction and is the most transparent and replicable (all data is shown in Figure 3).

We would also like to point out, that the first method (regression with interaction term) could be interpreted as a stratified method when looking at Figure 2: two regression lines (representing different uterine load categories) demonstrate a striking difference in their slopes, this visual representation is identical to one that would be obtained from the stratified approach (regression in two separate groups).

To clarify this issue, we have now enriched the manuscript text with interpretations of beta coefficients (obtained from the current regressions with interaction term), which are equivalent to those that would be obtained from the stratified approach.

6. The simulation models are not required, as they don't contribute to the discussion, and confuse the issue. Please remove.

The use of simulation models is not unusual in preterm birth research[1-3]. We feel that inclusion of the simulation model is a strength and a novel feature of our study, allowing us to use a classical hypothesis testing framework - a cornerstone of the scientific method. Furthermore, we believe that the simulation model improves the clarity of the results. We agree that the importance of the simulation model was understated in the original version of the manuscript, thus we highlighted it and provided a broader context in the Supplementary file. For additional transparency we have also provided the link to the simulation code, which could also be useful for other researchers.

Results:

1. The authors state on page 9 (lines 7-9) that “A clear interaction pattern was detected (figu.2 panel 1)”, however, judging from the figure this interactive effect is not evident. Is there a measure of statistical interaction, such as RERI or Synergistic index that the authors could quote to substantiate this claim? Rather, this seems like effect measure modification, as the slopes of the lines are approximately the same with different intercepts. For panel 2, the interactive effect is clear.

We thank the reviewer for the suggestion to use RERI or Synergistic index. In our view, this would be challenging to apply on a continuous data, so we refer the reader to the interaction term coefficients in Tables 1 and 2. We would also like to point out that the slopes of the lines (in Figure 2, Panel 1) differ approximately threefold: higher uterine load (twins) modulates the effect of maternal height on gestational age from 0.131days/cm (singletons) to 0.383 days/cm (0.131 + 0.252, twins). We would also like to note that our choice of coding for uterine load groups (0 and 1) enables the most straightforward interpretation of interaction results (Tables 1,2).

2. Tables 1 and 2 should not be in the results section text; rather they should be on a separate page each at the end of the manuscript. In fact, both tables could be presented as a single table.

Note that we have followed the BMJ instructions for authors that state "Tables should be in Word format and placed in the main text where the table is first cited."

3. Model coefficients are statistically significant; however, this is likely due to large sample size, meaning that the models are over powered. Please comment on the magnitude of the coefficient estimate. As an example, for table 1, “the independent effect of maternal height causes a 0.13 days increase in gestational age with every increase in maternal height unit, whereas the independent effect of increased uterine load causes an expected reduction of 69 days.” Please rephrase the results section with a focus on the magnitudes, rather than statistically significant p values. Please remove simulation analysis results.

Thank you for this great advice that will improve interpretation of the results. We have revised the text accordingly. Regarding simulation analysis results - we refer to the earlier comment (Methods 6).

Discussion:

1. “iatrogenic uterine stretch” implies that the uterus was stretched by the attending physician or due to treatment. Please correct this sentence.
Please note that in this sentence iatrogenic uterine stretch referred to experimental stretch study with pregnant nonhuman primate with inflation of catheter balloon. We have changed the word to “experimental” to make it clearer.

2. Please refer to specific study findings when stating “our results support ...”. Which of the study findings support these claims?
We have now clarified these sentences, thank you.

3. Sensitivity analysis mentioned on page 12 (line 18) was never explained in the methods. Also, the study population consisted of first pregnancies, so where did the authors acquire these additional pregnancies? Please remove this sentence or include it in the methods and results. We have now removed this sentence as suggested. Thank you.

4. Please temper the language on page 12 (line 22). “our results implicate candidate genes and proteins from inflammation mediated biochemical pathways...” The results from this study demonstrate an association, which may or may not be used to further research on predicting expected gestational duration.

We have now rewritten the quoted statement and included stronger references [4-7] that could support our claim. Our study supports a biomechanical pathway inducing labor onset, which is also supported by other experimental studies in nonhuman primates[8] and rodents[9]. Taken together, this body of knowledge could be used in genome-wide association studies to select a subset of candidate genetic markers, thus increasing the power of such studies. We have now toned-down the quoted statement and included stronger references that could support our claim.

Reviewer: 2

Reviewer Name: José Derraik

Institution and Country: • Liggins Institute, University of Auckland, Auckland, New Zealand • Uppsala University, Uppsala, Sweden

Competing interests: None declared.

Feedback:

I have enjoyed reading this interesting and well-written study that aims to examine a likely factor underpinning the previously described inverse association between maternal height and gestational length. This manuscript also highlights once again that the Swedish Birth Register is a powerful and valuable resource.

While I think this study is valuable and adds to the literature on this topic, I do have some questions regarding the manner in which the data were handled and analyzed.

Thank you for your supportive comments highlighting the strengths and novelty of our study.

Main concerns:

1. In one of its key comparisons, the authors contrast singleton and twin pregnancies. However, I don't think this comparison can be made to assess a possible association between uterine load and pregnancy length. Twin pregnancies are quite different from singleton ones. Most importantly, there is experimental evidence from animal studies showing that the growth trajectory (and to some extent pregnancy length) in twin pregnancies is likely determined very early in gestation (possibly even in the periconceptional period). For example, Hancock et al. showed that the surgical removal of a twin in sheep pregnancy did not lead to birth weight similar to that of singletons, while the length of gestation of such reduction pregnancies more closely approximated that of twin pregnancies rather than that of singletons (Hancock et al. *J Physiol* 2012;590:1273-84). As a result, I question the validity of singleton and twin pregnancies as surrogate categories for uterine load.

The reviewer makes an interesting point, which we now included in the discussion section by citing the Hancock study. At the same time, we would like to make two observations:

1) The quoted study (Hancock et al.) does not provide strong support for the reviewer's concerns. A randomly selected fetus was not removed, but only killed and left in the womb. Also, the control group (sham procedure) was not a perfect control as it differed from procedure group in other aspects - no KCl injection. As mentioned by Hancock et al.: "In late gestation sheep, complete surgical removal of one twin has no effect on gestation length of the remaining co-twin, whereas in pregnancies in which one twin has a cord ligation resulting in death and is left in utero, gestation length is decreased (Rueda et al. 1995)". We interpret Hancock's results with caution due to the inflammatory load imposed by a dead and decomposing fetus in the amniotic cavity.

2) Twin pregnancies are the closest surrogate for large uterine load that one can expect to have in humans; invasive experiments, like intrauterine balloon, would be unethical in pregnant women. We have checked the largest Scandinavian Medical Birth Registry for potential proxies to uterine load, and concluded that twin pregnancy was the best available candidate (other than LGA, which is discussed below).

2. Similarly, I see issues with the comparison between LGA and SGA pregnancies. SGA classifies most pregnancies likely to involve disordered fetal growth, which may result from a

wide range of underlying conditions (possibly undiagnosed) (Lee et al. Pediatrics 2003;111:1254-61). In particular, in this cohort solely of Nordic mothers and their offspring, it is unlikely that there were many SGA births resulting strictly from short parental stature. Thus, I think it is reasonable to assume that pregnancies leading to SGA births would unlikely constitute a group of health pregnancies for a valid comparison. On the other hand, a number of LGA births might have also resulted from undiagnosed cases of gestational diabetes mellitus.

We appreciate that the reviewer is concerned about extreme metabolic and vascular conditions that could be enriched in the two groups.

On the one hand, we would like to note that our study was less conservative in defining SGA and LGA (7% and 5%, respectively; 1.5 SD threshold) in comparison to the less inclusive medical definitions (2% and 1.7%, respectively; 2.0 SD threshold). Thus, our data is relatively depleted of intrauterine growth restriction and gestational diabetes.

On the other hand, we believe that reviewer's concern (two groups differing not only in fetal size, but also rates of various medical conditions) would more apply to the difference in means between the groups but not the slope (sensitivity of gestational age to maternal height). The qualitative result of our study is an observation that the slopes differ (i.e., there is an interaction).

Finally, it is important to look at our study design as a whole, i.e., to consider that two independent designs were used to model large-vs-small uterine load: twin pregnancies and LGA pregnancies.

- An important factor that the authors do not seem to have accounted for is maternal BMI. There is extensive evidence that maternal obesity is associated with an increased risk of post-term birth (e.g. Heslehurst et al. *Obes Rev* 2017;18:293-308). Conversely, data from the Swedish Birth Register also showed that maternal obesity is associated with an increased risk of preterm delivery (Cnattingius et al. *JAMA* 2013;309:2362-70). Thus, maternal BMI is an important confounding factor that needs to be accounted for in the analyses, as it may reduce or magnify the observed associations.

To make sure that reviewer's concerns are addressed, we ran sensitivity analyses, where BMI is restricted to a narrow window of $>22 \text{ kg/m}^2$ and $<24 \text{ kg/m}^2$ (± 1 unit around the study median, 23 kg/m^2), thus dramatically reducing the variation in BMI. The qualitative outcome did not change (Figure and Table below): there is a significant difference in the two slopes, one modeling gestational age dependence on maternal height in large uterine load (LGA) and the other - in small uterine load (SGA), gestational age being the most sensitive to maternal height in the higher uterine load group. The slope in SGA group is 0.1 days/cm, the slope in LGA group is 0.2 days/cm. Similar findings were observed in the sensitivity analysis using twin vs singleton pregnancies (not shown here). Thus we would doubt that BMI introduced the reported interaction pattern.

	Beta	SErr	Pvalue
Intercept	263,0	3,49	<1E-150
Maternal height (MH)	0,107	0,021	4E-7
Uterine load (UL)	-16,02	5,62	0,004
Interaction (MH x UL)	0,081	0,03	0,015

Uterine load (UL) was coded as "0" when SGA and "1" when LGA.

4. The authors have adopted some rather arbitrary thresholds without clear justifications for them. These include:

- non-standard definitions of SGA and LGA (page 6, lines 13-16).
There are a number of SGA/LGA definitions, some use percentiles; some use standard deviations (SD). SD-based definitions often use two thresholds: +/- 1.5 or +/- 2.0 SD [10]. We chose to use less conservative threshold in order to increase the sample size and limit any potential effects of SGA/LGA-associated comorbidities.-In two new sensitivity analyses described in our response to Reviewer 2 (Section "Suggestions" below), we show that it does not matter how you define the SGA and LGA - we still get the same qualitative results (i.e., interaction pattern with different slopes).
- the exclusion of mothers with BMI less than 15 kg/m² (may be better to exclude all underweight mothers, i.e. BMI <18.5 kg/m²)
This filter is mainly designed to catch erroneous data (maternal height) entries and not to represent the biological importance of BMI.
- what is the rationale for the 2.75 SD threshold? (page 7, line 12)
For the demonstration of effects at the extreme ends of the scale, we needed the largest SD threshold that would still leave at least 20 observations in the smallest group.
- mothers at the upper end of the BMI spectrum were also excluded, but why >45 kg/m²?
We chose this threshold arbitrarily to exclude data (maternal height) entries with a higher probability of handwriting and data entry (into the computer) errors.
- An intra-individual difference of 10 cm between height measurements is somewhat extreme. I am not sure why such a high threshold was adopted, as it suggests that the measured height of a number of mothers included in the study is likely to be rather inaccurate. This is concerning since maternal height is key for this study.

This threshold was used to catch large but infrequent errors in maternal height data (handwriting or subsequent typing into the computer). This filter affected only a very small number of pregnancies. In fact, a realistic estimate of true-error rates in maternal height data can be empirically calculated from twin pregnancies in the raw (preQC) data file: we would expect that maternal height would have the same value in each of the two data entries representing twins from the same pregnancy. We observe discordant height values in only 0.2% of twin pregnancies and discordant values with maternal height difference >10cm in only 0.03% of twin pregnancies.

The discussed 10 cm threshold used to clean height data based on all pregnancies of the same mother could have been smaller; however, that would have come at a cost of a high false-positive rate in calling errors, as mothers with recorded pregnancies at their young and mature age would be expected to have small height differences quite frequently.

- Page 6, lines 1-2 – This statement is vague and needs proper definition as well as adequate justification.
We removed this sentence, thank you.

Suggestions:

In light of the above, I would suggest tackling the key question differently. For example:

1. Focusing on singleton AGA pregnancies using birth weight SDS as a continuous variable. Also, possibly running stratified analyses, maybe looking at those $>-2SD$ but $<-1SD$ vs $>+1SD$ but $<+2SD$

Per the reviewer's suggestion, we ran both sensitivity analyses. The results are presented in Figures and Tables below. No qualitative change was observed from previous results.

Suggested sensitivity analysis 1: singleton AGA pregnancies (in a range of $>-2.0SD$ and $<2.0SD$ from the mean birthweight Z-score); linear regression with interaction term, using birth weight Z-score as a continuous variable.

	Beta	SErr	Pval
Intercept	253,8	0,429	$<1E-150$
MH	0,157	0,003	$<1E-150$
BWzs	-4,80	0,491	$1.4E-22$
MHxBWzs	0,024	0,003	$1.3E-16$

MH is maternal height. BWzs - birthweight Z-score.

To facilitate the interpretation of the beta coefficients in the table above, we provide an explanatory figure with four strata of birthweight Z-scores. Increasing birthweight Z-score increases the sensitivity of gestational age to maternal height.

Suggested sensitivity analysis 2: singleton pregnancies; linear regression with interaction term, using categorical uterine load variable, where SGA is defined as BW Z-score $>-2SD$ but $<-1SD$, and LGA is defined as BW Z-score $>1SD$ but $<2SD$.

	Beta	Serr	Tval	Pval
Intercept	261,1	1,026	254,6	$<1E-150$
MH	0,119	0,006	19,3	$9,4E-83$
UL	-12,46	1,69	-7,367	$1,8E-13$
MHxUL	0,062	0,01	6,20	$5,7E-10$

Uterine load (UL) was coded as "0" when SGA and "1" when LGA. MH is maternal height.

2. While the authors have considered only the first pregnancy for each mother, as the study covers a 15-year period, I think looking at siblings would add considerable value to this study. While it would be necessary to adjust for birth order, it would still be quite interesting to examine whether the reported association persists when one examines gestational length among siblings that were discordant for birth weight. The analyses could also be carried out on twin pregnancies separately.

We more than agree with the reviewer. As mentioned in the original manuscript text, the second, the third and further pregnancies of the same mother show much weaker interaction pattern, or no pattern at all. We find it to be a very interesting observation; it gives new insights into the hypothesis of uterine distention and thus will be included in a separate paper. We feel that irreversible physiological changes (remodeling) during the pregnancy could explain this phenomenon (biomechanical stresses to the uterus on the second and third pregnancy are likely different than on a first pregnancy).

Other comments:

1. Page 5, line 8 – I suggest replacing "kept" with "included".
We have modified as suggested.
2. Page 5, line 10 – Similarly, replace "removed" with "excluded".
We have modified as suggested.
3. Page 7, line 5 – Delete "a"
We have modified as suggested.
4. It is important that the authors include a paragraph commenting on the study's limitations.
We have now included study limitations in the Discussion, including the ones suggested by the Reviewer.
5. The references will also need to be edited to ensure a consistent formatting.
Thank you. We have modified as suggested.

References

1. Juodakis J, Bacelis J, Zhang G, et al. Time-Variant Genetic Effects as a Cause for Preterm Birth: Insights from a Population of Maternal Cousins in Sweden. *G3 (Bethesda)* 2017;7(4):1349-56. doi: 10.1534/g3.116.038612
2. Snowden JM, Basso O. Causal inference in studies of preterm babies: a simulation study. *BJOG* 2018;125(6):686-92. doi: 10.1111/1471-0528.14942

3. Suarez EA, Landi SN, Conover MM, et al. Bias from restricting to live births when estimating effects of prescription drug use on pregnancy complications: A simulation. *Pharmacoepidemiol Drug Saf* 2018;27(3):307-14. doi: 10.1002/pds.4387
4. Loudon JA, Sooranna SR, Bennett PR, et al. Mechanical stretch of human uterine smooth muscle cells increases IL-8 mRNA expression and peptide synthesis. *Mol Hum Reprod* 2004;10(12):895-9. doi: 10.1093/molehr/gah112
5. Sooranna SR, Engineer N, Loudon JA, et al. The mitogen-activated protein kinase dependent expression of prostaglandin H synthase-2 and interleukin-8 messenger ribonucleic acid by myometrial cells: the differential effect of stretch and interleukin-1{beta}. *J Clin Endocrinol Metab* 2005;90(6):3517-27. doi: 10.1210/jc.2004-1390
6. Sooranna SR, Lee Y, Kim LU, et al. Mechanical stretch activates type 2 cyclooxygenase via activator protein-1 transcription factor in human myometrial cells. *Mol Hum Reprod* 2004;10(2):109-13.
7. Terzidou V, Sooranna SR, Kim LU, et al. Mechanical stretch up-regulates the human oxytocin receptor in primary human uterine myocytes. *J Clin Endocrinol Metab* 2005;90(1):237-46. doi: 10.1210/jc.2004-0277
8. Adams Waldorf KM, Singh N, Mohan AR, et al. Uterine overdistention induces preterm labor mediated by inflammation: observations in pregnant women and nonhuman primates. *Am J Obstet Gynecol* 2015;213(6):830 e1-30 e19. doi: 10.1016/j.ajog.2015.08.028
9. Lye SJ, Mitchell J, Nashman N, et al. Role of mechanical signals in the onset of term and preterm labor. *Front Horm Res* 2001;27:165-78.
10. . In: Rasmussen KM, Yaktine AL, eds. *Weight Gain During Pregnancy: Reexamining the Guidelines*. Washington (DC), 2009.

VERSION 2 – REVIEW

REVIEWER	Abdool Yasseen Children's Hospital of Eastern Ontario - Research Institute Ottawa, Ontario, Canada
REVIEW RETURNED	17-Sep-2018

GENERAL COMMENTS	Dear authors, Thank you for addressing my concerns over excluded populations. I am happy that the results do not change drastically. While I do agree there is a time an place for simulation models, I believe the authors have made their case with the available empirical data, and the simulation is not needed. Further, in a medical journal such as BMJ Open, this information is not generally of interest to the reader. I would like to see this removed for clarity before I would agree to publication. Other than this small issue, I have no other concerns about the manuscript. Well done. Many thanks for allowing me to review,
---

REVIEWER	José Derraik Liggins Institute, University of Auckland, New Zealand
REVIEW RETURNED	25-Sep-2018

GENERAL COMMENTS	I commend the authors for their efforts in addressing the reviewers' comments, which have adequately addressed my concerns.
---

	However, I think it would be informative to the readers if the exploratory analyses (referred to as "Suggested sensitivity analyses 1 and 2" in the rebuttal document) were briefly referred to in a separate paragraph at the end of the Results section. Personally, I find these additional analyses useful. The respective tables and figures could be provided in a supplementary file. Other minor comments:  • p.7, line 17 "A birth concludes gestation" - this statement doesn't make sense to me. • Figure 1 - Best to include the units for BMI in the figure. I would also state in the top box that you are talking about the Swedish Birth Register. • Figure 3 - It would be helpful if "Risk" was explained in the figure legend. It is not immediately clear as to what it refers to. • Supplementary Figures - As these are stand-alone figures, I suggest include the full citation of the study being referred to as a footnote (e.g. Marsal et al. in Sup Fig 2). • Please ensure that the terminology is consistent. For example, in the Sup Figs y axes you have "birthweight" and then "birth weight"; Strenghts and.. (p.3) "registry" vs "register" throughout.
--	--

VERSION 2 – AUTHOR RESPONSE

Reviewer 1

Reviewer Name: Abdool S. Yasseen III

Institution and Country: Children's Hospital of Eastern Ontario - Research Institute, Ottawa, Ontario, Canada

Competing interests: None declared

Dear authors,

Thank you for addressing my concerns over excluded populations. I am happy that the results do not change drastically. While I do agree there is a time and place for simulation models, I believe the authors have made their case with the available empirical data, and the simulation is not needed. Further, in a medical journal such as BMJ Open, this information is not generally of interest to the reader. I would like to see this removed for clarity before I would agree to publication. Other than this small issue, I have no other concerns about the manuscript. Well done.

Thank you for the feedback. We have now removed the paragraph describing simulation model and all other references to it.

Reviewer: 2

Reviewer Name: José Derraik

Institution and Country: • Liggins Institute, University of Auckland, Auckland, New Zealand • Uppsala University, Uppsala, Sweden

Competing interests: None declared.

I commend the authors for their efforts in addressing the reviewers' comments, which have adequately addressed my concerns. However, I think it would be informative to the readers if the exploratory analyses (referred to as "Suggested sensitivity analyses 1 and 2" in the rebuttal document) were briefly referred to in a separate paragraph at the end of the Results section. Personally, I find these additional analyses useful. The respective tables and figures could be provided in a supplementary file.

Thank you for the feedback. We have now incorporated sensitivity analyses in the manuscript and the supplement as suggested.

Other minor comments:

1. p.7, line 17 "A birth concludes gestation" - this statement doesn't make sense to me. As part of changes requested by Reviewer 1, the whole section is now removed.
2. Figure 1 - Best to include the units for BMI in the figure. I would also state in the top box that you are talking about the Swedish Birth Register. The Figure 1 now includes requested modifications.
3. Figure 3 - It would be helpful if "Risk" was explained in the figure legend. It is not immediately clear as to what it refers to. The Figure 3 now contains a clearer column name: "Preterm birth risk"
4. Supplementary Figures - As these are stand-alone figures, I suggest include the full citation of the study being referred to as a footnote (e.g. Marsal et al. in Sup Fig 2). The citation is now included at the bottom of the page.
5. Please ensure that the terminology is consistent. For example, in the Sup Figs y axes you have "birthweight" and then "birth weight"; Strenghts and.. (p.3) "registry" vs "register" throughout. The manuscript and the supplement were checked for the described inconsistencies and modified accordingly. Thank you for noticing this.